# Pathogenic *Leptospira* Species Are Present in Urban Rats in Sydney, Australia

**DOI:** 10.3390/microorganisms11071731

**Published:** 2023-07-01

**Authors:** Miguel A. Bedoya-Pérez, Mark E. Westman, Max Loomes, Nga Yee Natalie Chung, Benjamin Knobel, Michael P. Ward

**Affiliations:** 1Brain and Mind Centre, The University of Sydney, Sydney, NSW 2050, Australia; miguel.bedoyaperez@sydney.edu.au (M.A.B.-P.); max.loomes@sydney.edu.au (M.L.); 2School of Psychology, Faculty of Science, The University of Sydney, Sydney, NSW 2006, Australia; 3Sydney School of Veterinary Science, The University of Sydney, Sydney, NSW 2006, Australia; mark.westman@sydney.edu.au (M.E.W.); nchu3419@uni.sydney.edu.au (N.Y.N.C.); 4Elizabeth Macarthur Agricultural Institute (EMAI), Woodbridge Road, Menangle, NSW 2568, Australia; benjamin.knobel@dpi.nsw.gov.au

**Keywords:** antibodies, diagnosis, leptospirosis, One Health, *Rattus rattus*, *Rattus norvegicus*, urban rodents, veterinary science

## Abstract

Leptospirosis is an emerging disease among people and dogs in Sydney, Australia. However, the routes of *Leptospira* transmission in these cases, and in particular the possible role of rats as reservoirs of infection in Sydney, are unknown. Rats were collected within the City of Sydney Council area and their kidneys were tested for pathogenic *Leptospira* DNA by real-time (q)PCR. A subset of rats also had qPCR testing performed on whole blood and urine, and Microscopic Agglutination Testing (MAT) that included a panel of 10 *Leptospira* serovars from nine different *Leptospira* serogroups was performed on a subset of serum samples. Based on qPCR testing, the proportion of rats with *Leptospira* DNA in their kidneys was 9/111 (8.1%). qPCR testing of blood samples (*n* = 9) and urine (*n* = 4) was negative. None of the 10 serum samples tested MAT positive. A primary cluster of qPCR-positive locations was detected based on six infected rats, which partially overlapped with a previously identified cluster of canine leptospirosis cases in Sydney. These findings suggest that rats in Sydney might play a role in the transmission of leptospirosis to dogs and people. Further testing of rats in Sydney and investigation into other possible wildlife reservoirs of infection and environmental sources of leptospires are needed.

## 1. Introduction

Leptospirosis is contracted by about one million people annually, with 60,000 dying from the disease [1]. A cork-screw-shaped bacterium, *Leptospira*, causes the disease. There are 18 species of *Leptospira* and 200 different ‘serovars’, or genetic variants, of the bacterium [2]. Most species of *Leptospira* (such as *L. interrogans,* which causes most human deaths) are pathogenic and are also highly transmissible [3]. According to Costa et al. [4], brown rats (*Rattus norvegicus*) are the main reservoir for *L. interrogans* in urban areas. Humans and domesticated animals are usually infected by *Leptospira* via direct contact with an infected rodent (i.e., bites or scratches) or indirectly through contamination of water sources by urine from an infected rodent [5]. Symptoms in humans include sore muscles, migraines, fever, vomiting and in more severe cases, Weil’s disease or meningitis [5], which can cause death if not treated. Although an infection can be effectively treated by antibiotics such as amoxicillin [6], leptospirosis remains a problem and disproportionately affects large, densely populated cities in developing nations [7,8].

Even though leptospirosis is prevalent in lower socioeconomic status regions, it can also arise in other areas. In Australia, traditionally the highest number of human cases of leptospirosis occur in the state of Queensland, with 330 human cases recorded between 2015–2018 [9]. There have also been recent cases of leptospirosis among people in New South Wales (NSW), Australia. The first recent outbreak occurred on a rural NSW mixed-berry farm in 2018, when 84 workers showed symptoms of leptospirosis. Of these workers, 22 were hospitalized. Further tests identified *L. borgpetersenii* serovar Arborea and *L. interrogans* serovar Zanoni as the primary causes of illness [10]. Evidence of on-site rodent activity prompted the capture and testing of 13 mice, and 3 mice were confirmed to be *Leptospira* carriers [10]. Serovar Arborea is found in rats and mice worldwide and was first detected in Australia in 1998 [11]. In 2021, a rare urban-acquired case of human leptospirosis was reported in a worker at a Sydney golf course [12]. PCR testing of water and soil samples collected on site were positive for pathogenic *Leptospira* spp. (manuscript in preparation).

A 2008 Australian serosurvey of dogs reported six seropositive dogs from a Sydney shelter (Yagoona), of which three dogs (50%) had titers against *L. interrogans* serogroup Icterohaemorrhagiae, and a local rat problem was hypothesized [13]. However, canine leptospirosis had not been reported in the Sydney dog population since 1976. In 2017, after 41 years of seeming leptospirosis quiescence, a series of canine cases started to emerge in Sydney. Of these cases, five were from the inner-city suburbs of Glebe and Surry Hills [14]. All five cases died, well above the usual canine fatality rate of 20–48% [15]. By 2020, the total number of leptospirosis cases in dogs in the Greater Sydney area reached 17, with 13 of those within the City of Sydney Council; only 2 dogs survived the infection. Overall, the fatality rate in the 17 canine leptospirosis cases reported between 2017–2020 was 88% (15/17). Most cases tested were positive for serovar Copenhageni, with one dog positive for serovar Hardjo and one for serovars Australis and Robinsoni (all serovars belong to *L. interrogans*) [14]. It was suggested that most canine infections occurred through urine transmission or the consumption of rodents. Building works nearby and around Sydney were hypothesized to have increased the number of rodents in the area and allowed them access to waterways. As a public health precaution, all dogs within a three-kilometre radius were advised to be vaccinated against serovar Copenhageni (Protech^®^ C2i, Boehringer Ingelheim Animal Health Australia).

While vaccination protects dogs against particular serovars capable of causing canine leptospirosis, vaccines are serovar-specific; thus, vaccinated dogs are not protected against other serovars that might be present in the rodent population. Protech^®^ C2i is the only registered vaccine in Australia, so vaccinated dogs are still susceptible to other serovars including serovar Zanoni [9]. A restricted-use vaccine (AUSLEPTO, Tre’idila Biovet), developed to help protect dogs against disease caused by serovar Australis, is also available in Queensland and the Northern Territory [9] and, more recently, in the local government area of Shoalhaven NSW. At least one fatal case of canine leptospirosis has been reported in a dog vaccinated with Protech^®^ C2i nine months prior. However, the causative serovar could not be determined, likely due to insufficient time for seroconversion [14].

To further determine the possible role of rats in canine and human leptospirosis in Sydney, Australia, it is necessary to investigate whether rodents are reservoirs of *Leptospira* infections. In this study, our aim was to estimate the prevalence of pathogenic *Leptospira* spp. in the urban rat population.

## 2. Materials and Methods

### 2.1. Live Trapping

Using ArcGIS 10.5 (Redlands, CA, USA) [16], a grid of 167 quadrats (0.25 km^2^) was overlaid on the area within the Council of the City of Sydney boundaries. The size of the quadrats was selected based on previously reported maximum home ranges for urban rats [17]. From the 167 quadrats, 15 were initially selected by systematic random sampling to ensure the inclusion of areas varying in waste management schedules and green areas with dog on-leash and off-leash policies.

Trapping occurred on eight consecutive sessions from 23 June to 13 August 2020. Each trapping session ran for six nights, including three nights of acclimatization and three nights of trapping, and occurred in two randomly selected quadrats from the 15 previously selected. Due to restricted access, or inability to secure traps in one of the pre-selected quadrats, the trapping site was moved east to the adjacent quadrat. Additionally, an extra randomly selected quadrat was added to the sampling schedule, producing 16 trapping sites in total. During each trapping session, ten live cage traps (40.64 cm L × 12.7 cm W × 12.7 cm H; model 602; Tomahawk, WI, USA) were deployed in each quadrat, distributed as evenly as allowed by the urban landscape.

Traps were baited before sunset using a mixture of peanut butter, rolled oats and honey. During the three nights of acclimatisation, traps were locked open and baited to allow the rats to familiarise themselves with the traps. On the fourth day, traps were set before dusk and checked around dawn. Once captured, animals were transported to the University of Sydney (UoS), where they were euthanised using intraperitoneal sodium pentobarbital. Urine, blood and kidney samples were immediately collected post-mortem and stored at −80 °C. Urine was collected by cystocentesis and stored in sterile cryotubes. Blood was collected by cardiac puncture and transferred to ethylenediamine tetra-acetic acid (EDTA) tubes and plain tubes. Plain tubes were allowed to stand for at least 10 min, with serum harvested using a sterile pipette and transferred to plain sterile cryotubes prior to storage. Kidneys were extracted post-mortem and stored whole in sterile cryotubes.

### 2.2. Multi-Catch Rodent Station Samples

Under a pest management contract between the Council of the City of Sydney and Flick Anticimex Pty Ltd., 20 to 60 multi-catch rodent stations have been deployed outdoors in public places and on Council owned land since October 2019. Deployment has been non-random and guided by strategic pest management priority zoning. The Flick SMART Multi-Catch killing rodent station is an internationally patented rodent trap design [18]. The trap consists of a trigger mechanism that instantaneously kills the caught animal with an electric current. The trap has a built-in programmable computer with a SIM card, enabling it to communicate via the mobile network when activated. This trap can catch multiple animals (eight maximum) before it needs to be serviced.

The trap stations are baited with barbeque grease and commercially available rodent attractants. Flick Anticimex Pty Ltd. staff check multi-catch rodent stations regularly. From 10 September 2020 to 7 May 2021, carcasses were collected from these stations and transported to the UoS for kidney tissue sample extraction by post-mortem. Urine and blood samples were not collected from these animals due to tissue degradation between death and carcass collection. Carcasses were received within 4 h of collection from traps. However, some carcasses could have remained in the traps for up to a week before collection.

### 2.3. Leptospira Detection by Real-Time PCR Testing

DNA from a small amount of thawed fresh-frozen kidney tissue (<35 mg), 200 µL anticoagulated EDTA blood or 200 µL of pelleted urine was extracted using either the QiaAMP DNA Mini Kit or the Qiagen DNeasy Blood and Tissue Kit (Qiagen Inc., Toronto, ON, Canada) according to the manufacturer’s instructions. DNA was then stored at −80 °C until batch PCR testing was performed. DNA extraction and PCR testing were performed at two sites using two different published PCR assays.

At the UoS, at least three aliquots of extracted kidney DNA were analyzed using a real-time (q)PCR assay, which targets a 242 bp fragment of the *lipL32* gene of pathogenic *Leptospira* species [3]. More than 50% of aliquots needed to test positive (i.e., at least 2 out of 3 aliquots) for a sample to be classified qPCR positive. Whenever inconclusive results were obtained, another three aliquots of kidney DNA were tested. Again, more than 50% of aliquots needed to test positive (i.e., at least 4 out of 6 aliquots in total) for a sample to be classified qPCR positive. The reactions were performedusing a StepOnePlus^TM^ Real-Time PCR System. Each 20 µL reaction contained 10 µL TaqMan Universal PCR Master Mix, 0.4 µL probe (5 µM), 0.6 µL forward and reverse primers (10 µM each), 7.7 µL molecular grade water (MGW) and 1.3 µL extracted DNA. Cycling conditions consisted of 50 °C for 2 min, followed by 95 °C for 10 min, followed by 45 cycles of amplification (95 °C for 15 s and 58 °C for 60 s). This has been shown to be highly sensitive and specific, with a lower limit of detection (LOD) of 0.013 fg/µL [3]. DNA extracted from a pathogen-free male *R. norvegicus* laboratory rat (Wistar) was used to create positive (spiked) and negative controls (unspiked). Positive controls were spiked with 5 ng of the synthesised known 242 bp fragment, and negative controls were unspiked. Additionally, a positive control consisting of only 5 ng of the synthesised known 242 bp fragment and a blank were also included in each run. Controls were included in triplicate to determine if rat genomic DNA would interfere with the detection, either by blocking the amplification of the fragment or by erroneous amplification of rat DNA by the probes used. Neither of those cases was detected in our assay, with comparable Ct levels for both types of positive (i.e., spiked and fragment) and negative controls (i.e., unspiked and blank).

At the Elizabeth Macarthur Agricultural Institute (EMAI), aliquots of extracted DNA were analyzed using a different qPCR assay [3], which targets an 87 bp sequence of the *rrs* (16S) gene of pathogenic *Leptospira* species. The reactions were performed using a QuantStudio™ 5 Real-Time PCR machine. Each 20 µL reaction contained 10 µL TaqMan^TM^ Environmental Master Mix 2.0, 0.4 µL probe (5 µM), 0.6 µL forward and reverse primers (10 µM each), 6.4 µL MGW, and 2 µL extracted DNA. Cycling conditions consisted of 95 °C for 10 min, followed by 95 °C for 15 s and 60 °C for 60 s for 45 cycles. An in-house-designed *Leptospira* plasmid standard (the 87 bp *rrs* target from a positive field sample received at EMAI and cloned into a pCR2.1 TOPO™ vector, ThermoFisher, Waltham, MA, USA) was created and used for all qPCR runs as a positive control and to form a standard curve to quantify results. The correct sequence of the 87 bp plasmid insert was confirmed at EMAI prior to testing by NCBI BLASTn of the target. Eight replicates of the *Leptospira* plasmid DNA at 20, 2, 0.2, 0.02, 0.002 and 0.0002 fg/μL were used to calculate a LOD using GraphPad Prism software Version 4.02 (San Diego, CA, USA) with a nonlinear regression model using a sigmoidal dose-response. The 50% LOD for the assay was determined to be 0.026 fg/µL. This LOD was used to determine a positive or negative result. A positive result was determined when amplification in the sample well was above the LOD. A negative result was determined when no amplification occurred in the sample well, or when amplification was below the LOD (i.e., <0.026 fg/µL). A standard curve was created by running four known plasmid standard concentrations from a dilution series in duplicate, with acceptable run criteria defined as R^2^ > 0.98 and efficiency = 85%–110%. If the standard curve values fell outside of these criteria, the run was deemed to have failed and testing was repeated. Three negative controls (PCR reagents without template DNA added; PCR reagents with MGW added instead of template DNA; and an extraction process control) were also included in each qPCR run.

Some samples had DNA extraction performed at the UoS and qPCR testing performed at both the UoS and EMAI using the same aliquots of DNA. A sample with a positive qPCR result at either site was considered positive in these cases.

### 2.4. Microscopic Agglutination Testing

Microscopic Agglutination Testing (MAT) was performed on serum samples (when available) as per normal commercial testing protocols at EMAI. This included testing against 10 different *Leptospira* serovars (sv.), pertaining to 9 different *Leptospira* serogroups: Serogroup Ballum (sv. Arborea), serogroup Australis (sv. Australis), serogroup Canicola (sv. Canicola), serogroup Icterohaemorrhagiae (sv. Copenhageni and sv. Icterohaemorrhagiae), serogroup Grippotyphosa (sv. Grippotyphosa), serogroup Sejroe (sv. Hardjo), serogroup Pomona (sv. Pomona), serogroup Tarassovi (sv. Tarassovi) and serogroup Pyrogenes (sv. Zanoni). An agglutination result at 1/50 dilution or higher was considered a positive titre.

### 2.5. Spatial Analyses

Locations at which positive or negative samples were detected were mapped (ArcGIS Version 10.5) using recorded latitude and longitude coordinates. Cluster analysis was performed using spatial scan statistic with a Bernoulli (case-control) model. The data were scanned for clusters with high rates using a circular scanning window of 10% of the study area. The significance of the log likelihood statistic was determined using a Monte Carlo simulation with 999 permutations. Locations were also mapped based on the estimated concentration of pathogenic *Leptospira* DNA (fg/µL) in positive samples. To characterize clustering, Moran’s autocorrelation statistic (I) was calculated (ArcGIS Version 10.5 Spatial Analyst).

## 3. Results

### 3.1. Rodent Survey (n = 120)

From a total of 480 trap/nights (i.e., ten traps per site for three nights), 16 rats were captured by live trapping. Another 104 rat carcasses were obtained from Flick SMART Multi-Catch killing rodent stations. The demographics of the 120 rats are shown in Table 1.

There were 111 rats with kidney samples for *Leptospira* testing (9 of the 104 carcasses obtained from Flick SMART Multi-Catch killing rodent stations were too decomposed to obtain viable kidney samples). There were 85 rats available for mapping (a further 26 rats were from unknown locations and thus could not be included in the spatial analyses).

### 3.2. Leptospira Detection in Kidney Samples by qPCR Testing (n = 111)

Based on combined results from qPCR testing at both sites, the proportion of rats carrying *Leptospira* DNA was 9/111 (8.1%).

Kidneys from 28 rats (16 live traps and 12 carcasses) were collected and qPCR tested at the UoS. Of these samples, 6/28 (21%) were qPCR positive. The mean Ct of positive samples was 34.4 (Ct range 26.8–45.0).

Kidneys collected from all 111 rats were qPCR tested at EMAI. Of these samples, 6/111 (5.4%) were positive. Five of these six qPCR-positive kidney samples were also positive with qPCR testing at the UoS. The mean Ct of positive samples at EMAI was 26.8, and the mean concentration was 49.7 fg/µL (Ct range 25.3–28.9; concentration range 9.1–96.0 fg/µL). Another kidney sample was below the LOD and classified as qPCR negative (Ct 38.4, concentration 0.017 fg/L); this sample had previously tested qPCR negative at the UoS (Table 2).

Of the 111 rats, 61 also had kidney DNA extraction performed at the UoS and qPCR testing performed at EMAI. An additional two DNA samples were determined to be qPCR positive (Sample 1: Ct 36.6, concentration 0.057 fg/µL; Sample 2: Ct 39.98, concentration 0.036 fg/µL) (Table 2).

In total, 4/16 (25%) live-trapped rats were qPCR positive, and 5/95 (5.3%) rat carcasses from killing stations were qPCR positive. Live-trapped rats were more likely to test qPCR positive for *Leptospira* DNA than collected rat carcasses (*p* = 0.024; Fisher’s exact test).

All nine *Leptospira*-positive rats were male, 8/9 of which were brown rats, *R. norvegicus* (7.1%, *n* = 113), and only 1/9 was a black rat, *R. rattus* (14.3%, *n* = 7). There was no difference in prevalence between rat species (*p* = 1.0; Fisher’s exact test).

### 3.3. Leptospira Detection in EDTA Blood Samples (n = 9) and Urine Samples (n = 4) by qPCR Testing

Nine rats had anticoagulated EDTA blood samples, and four had urine samples, available for *Leptospira* qPCR testing. None of the EDTA blood or urine samples were qPCR positive. All EDTA blood and urine samples were collected from rats that tested qPCR negative with kidney testing.

### 3.4. MAT Testing on Serum Samples (n = 10)

Ten rats had serum available for MAT, collected from rats that tested qPCR negative with kidney testing. All ten rats tested antibody negative for all 10 *Leptospira* serovars.

### 3.5. Spatial Analysis of Leptospira Prevalence in the Rodent Population (n = 85)

The distribution of *Leptospira* qPCR-positive and qPCR-negative sampling locations is shown in Figure 1. A primary cluster of qPCR positive locations was detected. This consisted of eight locations in the north-central part of Sydney City (33.872845° S, 151.211910° E), radius 0.96 km, in which there were six qPCR-positive samples (0.74 qPCR-positive samples expected; observed/expected, 8.10; log likelihood ratio, 13.12794, *p* = 0.00011). No secondary clusters were identified. The distribution of *Leptospira* DNA concentrations were found to be clustered (Moran’s I = 0.2202, *p* < 0.0001), with higher concentrations within the area in which clustered qPCR-positive locations were identified.

## 4. Discussion

Brown rats (*R. norvegicus*) and one black rat (*R. rattus*) were found in this study to have pathogenic *Leptospira* DNA in their kidneys. Positive samples were clustered in the north-central part of Sydney City, with higher *Leptospira* DNA concentrations and a partial overlap with previous cases of canine leptospirosis identified. Although the viability of the *Leptospira* DNA material detected was not studied, and direct transmission was not demonstrable, these findings suggest that rats in Sydney might play a role in the transmission of leptospirosis to dogs and people.

Reports on global distribution of *Leptospira* infection in rats vary considerably based on geographic location, with prevalence as high as >80% reported in studies in Brazil, India, and the Philippines [20]. Results from the current study represent one of the few reports of pathogenic *Leptospira* spp. in rats in Australia, and the first report from Sydney. A large review reported a combined seroprevalence in Australia based on MAT of 0–1.7% and a combined prevalence based on molecular testing of 0–22% from three peer-reviewed studies [20]. The prevalence of positive kidney samples for leptospires in the current study (9/111, 8.1%) is therefore comparable to previously published estimates.

Australia is a large continent with greatly varying climate, and this certainly affects the epidemiology of *Leptospira* infection in carrier animals including rodents. From 70 non-native rodents trapped in four household/dairy properties in the Atherton Tablelands, North Queensland, leptospires were detected by conventional PCR testing of pooled organ samples (liver, spleen, kidney, and ileum) in 24/52 (46%) house mice (*M. musculus*), 4/17 (24%) black rats (*R. rattus*), and 0/1 (0%) brown rats (*R. norvegicus*). No rodents were MAT positive [21]. In 2017, 2/68 (2.9%) black rats on Christmas Island were reported positive for pathogenic *Leptospira* (*L. interrogans*) by conventional nested PCR testing of kidney samples, the first time *Leptospira* spp. had been identified in animals on this tropical Australian island located approximately 360 km south of Jakarta [22]. In a large serosurvey of a range of wildlife in more temperate southeastern Australia, 2/100 (2%) bush rats (*Rattus fuscipes*) were MAT positive (one to *L. interrogans* serovar Australis, one to *L. borgpetersenii* serovar Ballum), while 0/9 water rats (*Hydromys chrysogaster*), 0/10 black rats (*R. rattus*), 0/10 eastern swamp rat (*Rattus lutreolus*), and 0/4 broad-toothed rat (*Mastacomys fuscus*) were MAT positive [23]. An earlier study of animal reservoirs of leptospires in Northern Queensland reported a much higher prevalence in both native and non-native rodents than southeastern Australia, with a combined prevalence (based on a positive culture or MAT result) of 9/13 (69%) in brown rats (*R. norvegicus*), 36/108 (33%) canefield rats (*R. sordidus conatus*), 12/48 (25%) water rats (*Hydromys chrysogaster*), 22/130 (17%) house mice (*M, musculus*), 19/137 (14%) allied rats (*R. assimilis*), 49/411 (12%) black rats (*R. rattus*), 2/26 (8%) giant naked-tailed rats (*Uromys caudimaculatus*), and 2/252 (1%) naked-tailed rats (*Melomys* spp.) [24]. The authors of the southeastern Australia report presumed that the lower prevalence of *Leptospira* infection/exposure in their study compared to more tropical north Queensland was due to climatic differences [23]. Three of thirteen (23%) mice tested positive for *L. borgpetersenii* serovar Arborea in the outbreak of human leptospirosis on the NSW mixed-berry farm in 2018, but no rats were tested [10]. Sydney’s climate is classified as ‘humid subtropical’, and with the threat of a warming climate, the risk of leptospirosis transmitted from rats to dogs and humans is likely to increase [9].

A review of 145 *Leptospira* studies reported a higher *Leptospira* prevalence in *R. norvegicus* than other *Rattus* spp. [20]; this was not the case in the current study, although low numbers of *R. rattus* collected may have precluded statistical significance. Further rat sampling to investigate *Leptospira* prevalence in different rat species would be prudent to better understand the role of rats in *Leptospira* transmission in Sydney.

MAT of a subset of samples was unfortunately unable to identify the *Leptospira* serogroups present in rats in the current study. Serum from only 10/16 live trapped rats was able to be obtained, and all ten rats that were MAT negative were also qPCR negative for *Leptospira* with kidney testing. Blood sampling from the 104 rat carcasses was not possible due to body decomposition, which was a limitation of the study. Kidney samples from live-trapped rats were significantly more likely to test qPCR positive than rats collected from killing stations, likely due to sample DNA decomposition, considering up to one week might have passed between death and carcass collection. Although live trapping is much more laborious than collecting carcasses, with daily trap checks required for six consecutive days, future studies should focus on live rat trapping to minimize the possibility of false-negative qPCR results, facilitate blood collection and MAT, and increase the likelihood of successful bacterial culture to identify the *Leptospira* serovar/s present. In general, we found that the qPCR assays used (UoS and EMAI) were highly congruent, with only one sample classified as qPCR positive at EMAI but negative at the UoS, and one sample classified as qPCR negative at EMAI but positive at the UoS. These discordant results might have been due to differences in the sensitivities of the two qPCR assays [3,19]. Attempts have been made to design *Leptospira* serovar-specific PCR primers, but to date no such PCR assay exists [25].

A cluster of qPCR rat kidney samples was identified in the north-central part of Sydney City. This area is centered on the Sydney Central Business District. This highly urbanized area is dominated by office and retail infrastructure, but also contains moderately-sized parklands and recreational areas. Why a spatial cluster of *Leptospira*-positive rats was present in this area is unknown. A possible factor might be the urban landscape facilitating the movement of rats and contact structures that support *Leptospira* transmission. Sydney is also a city surrounded by water and containing many water features, both recognized risk factors for leptospirosis [26]. This finding has potentially important implications for rat management interventions from a One Health perspective. One Health is an all-encompassing term that considers animal, human, and environmental health, and particularly the interplay between all three aspects [27]. Further sampling of rats in this area of the city over time is needed to better understand the epidemiology of *Leptospira* in rats in highly urbanized areas.

It is important to note that there was some overlap between the cluster of *Leptospira* positive rats identified in the current study, and a 2019 cluster of canine leptospirosis cases identified in a previous study [28] (Figure 1). This provides some evidence that rats might be acting as reservoirs for the canine cases that have been and continue to be reported in Sydney. However, determining the serovars present in the rat population is required to infer a reservoir role. Ongoing serological and molecular surveillance of possibly infected dogs and rats in the Sydney area and bacterial culture for *Leptospira* organisms are therefore important. Since vaccines to prevent leptospirosis are serovar-specific, identification of common serovars by culture and genomic sequencing will also assist with vaccine development and other related preventative measures.

Other investigations of potential wildlife reservoirs of *Leptospira* infection and environmental sources of *Leptospira* contamination (in particular water reservoirs) are also needed to better understand the risks of direct and indirect transmission of leptospirosis in Australia. Few studies of possible wildlife reservoirs have been performed in Sydney. Serology testing of common brushtail possums (*Trichosurus vulpecula*) in urban Sydney identified two serovars of *Leptospira* in 9.6% (13/136) of individuals caught [29]. One serovar identified in two brushtail possums was likely serovar Arborea, one of the causes of the 2018 outbreak on the NSW mixed-berry farm (*L. interrogans* serovar Hardjo was identified in the other 11 brushtail possums) [10,29]. A serologic-based investigation of free-ranging eastern grey kangaroos (*Macropus giganteus*) captured in the Warragamba Catchment Area, Sydney, detected antibodies against *L. weilii* serovar Topaz in 47% (41/87) of the serum samples collected. Antibodies against no other *Leptospira* serovars were detected [30]. A serological survey of free-living platypuses (*Ornithorhynchus anatinus*) in the Wollondilly River, approximately 200 km south of Sydney, reported that 67% (14/21) were positive for antibodies against *Leptospira interrogans* [31]. Although kangaroos and platypuses might not be present in the Central Business District of Sydney, these studies highlight the types of surveillance studies needed to investigate the potential role of urban-adapted wildlife in Sydney as reservoirs of *Leptospira* infection including flying foxes (bats), red foxes, and birds (e.g., the Australian white ibis). Furthermore, environmental testing of water and soil samples in Sydney and elsewhere in Australia, which, to the authors’ knowledge, has not been reported in the peer-reviewed literature, should be performed to better understand other local risk factors for leptospires transmission.

## 5. Conclusions

This study reports the prevalence of pathogenic *Leptospira* DNA in the kidneys of wild rats caught in the City of Sydney Council area to be 8.1%. A cluster of six qPCR samples was identified in the Sydney Central Business District, with some overlap identified with a cluster of canine leptospirosis cases in 2019. Although not directly demonstrating causation, these results suggest that rats in Sydney might play a role in the recent transmission of leptospirosis to dogs and people in Sydney. Further research is required to determine the *Leptospira* serovars carried by rats, as well as other potential wildlife reservoirs of infection and environmental sources for leptospires. Human physicians in Sydney and elsewhere in NSW should consider leptospirosis in at-risk patients presenting with symptoms possibly attributable to infection with leptospires.

## Figures and Tables

**Figure 1 microorganisms-11-01731-f001:**
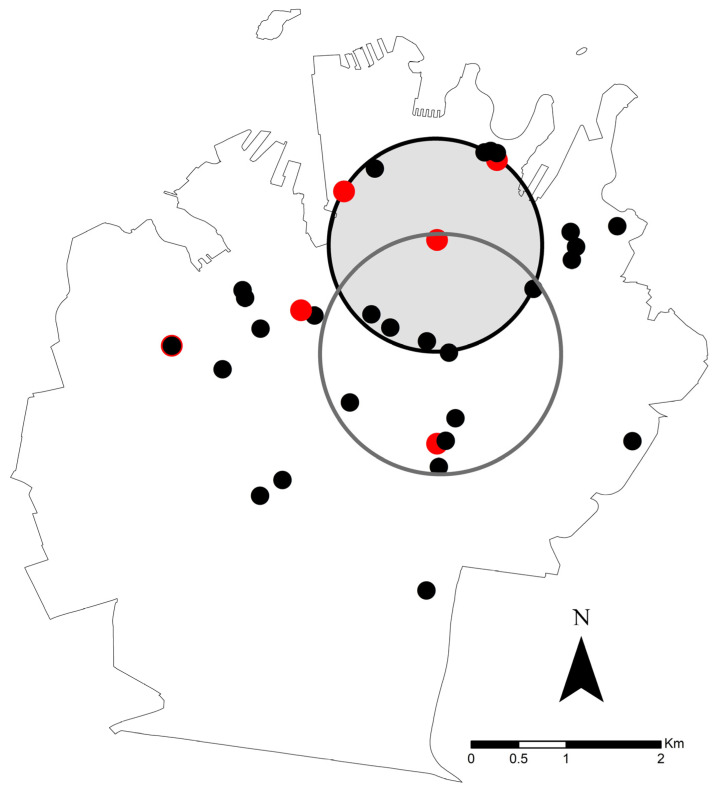
The distribution of *Leptospira* qPCR-positive (red) and qPCR-negative (black) rat sampling locations, and a primary cluster (grey-filled circle) of qPCR positive locations (six qPCR-positive samples; 0.74 qPCR-positive samples expected; observed/expected, 8.10; log likelihood ratio, 13.12794, *p* = 0.00011). For comparison, a previously identified cluster of canine leptospirosis cases in 2019 [19] is shown (open circle with grey border).

**Table 1 microorganisms-11-01731-t001:** Demographics of rat samples obtained through live trapping from 23 June to 13 August 2020, and Flick SMART Multi-Catch rodent station from 10 September 2020 to 7 May 2021. NA = not applicable.

		Captured				Mean Weight ± SE (g)			Reproductive State		
Source	Species	♂	♀	?	Total	♂	♀	?	♂	♀	?
Live trapping (*n* = 16)	Brown rats (*R. norvegicus*)	8	2	0	10	273.5 ± 24.9	213.5 ± 84.6	NA	2 Sub adults 6 adults	2 Non-breeding adults	
	Black rats (*R. rattus*)	5	1	0	6	140.6 ± 18.7	175 ± 0	NA	3 Juveniles 2 Adults	1 Non-breeding adult	
	Total	13	3	0	16						
Multi-Catch killing stations (*n* = 104)	Brown rats (*R. norvegicus*)	52	43	8	103	84.5 ± 11.1	77.7 ± 6.5	41 ± 5.7	49 Juveniles 3 Adults	26 Juveniles 17 Non-breeding adults	8 Adults
	Black rats (*R. rattus*)	1	0	0	1	100 ± 0	NA	NA	1 Juvenile		
	Total	53	43	8	104						
Total		66	46	8	120						

**Table 2 microorganisms-11-01731-t002:** Summary of results from qPCR testing of 120 rat samples. All qPCR testing of whole anticoagulated EDTA blood and urine was performed at the Elizabeth Macarthur Agricultural Institute (EMAI). UoS = the University of Sydney, +ve = positive (colored red for clarity), −ve = negative, NP = not performed, NA = not available.

Rat Samples for qPCR Testing	Kidney #1 (Extraction and qPCR Testing Both at EMAI)	Kidney #2 (Extraction and qPCR Testing Both at UoS)	Kidney #3 (Extraction at UoS, qPCR Testing of Extract at EMAI)	EDTA Blood	Urine
Live trapping (*n* = 16)					
*n* = 2	+ve	+ve	NP	NA	NA
*n* = 1	+ve	−ve	NP	NA	NA
*n* = 1	−ve	+ve	NP	NA	NA
*n* = 1	−ve *	−ve	NP	NA	NA
*n* = 3 ^#^	−ve	−ve	NP	−ve	−ve
*n* = 1 ^#^	−ve	−ve	NP	NA	−ve
*n* = 6 ^#^	−ve	−ve	NP	−ve	NA
*n* = 1	−ve	−ve	NP	NA	NA
Killing stations (*n* = 104)					
*n* = 3	+ve	+ve	NP	NA	NA
*n* = 9	−ve	−ve	NP	NA	NA
*n* = 2	−ve	NP	+ve	NA	NA
*n* = 59	−ve	NP	−ve	NA	NA
*n* = 22	−ve	NP	NP	NA	NA
*n* = 9	NA	NA	NA	NA	NA
***n* = 120**	6/111	6/28	2/61	0/9	0/4

* Inconclusive since sample below the limit of detection (LOD). ^#^ These 10 rats were MAT negative against 10 *Leptospira* serovars from nine different *Leptospira* serogroups.

## Data Availability

All data presented in this paper are available upon request.

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
