# Peer review of "Pathogenic Leptospira Species Are Present in Urban Rats in Sydney, Australia"

_microorganisms, 2023, doi:10.3390/microorganisms11071731_

Round 1

Reviewer 1 Report

Correct the manuscript  as the file attachment

Update the references

correct the English language

Author Response

Thank you to this Reviewer for very careful edits to our manuscript! They have all been attended to and are greatly appreciated. There was one suggested edit that we did not adopt, however, and we trust this Reviewer will understand our decision with a supporting explanation. We left ‘veterinary science’ as a key word, since we believe the subject matter is relevant to veterinarians, and our veterinary school has requested that we include this as a key word on any publications to assist with article visibility, particularly for veterinarians and other veterinary schools.

Reviewer 2 Report

1. Authors very well analyzed data from results; i.e., trapping methods, number of nights, qPCR, CT values, MAT serovar screening, etc

2. I don't like the live rats' sample size; 16 versus 104 carcasses. I understand that working in the field is quite different from lab work. But, I would not be surprised if removing carcass numbers in the denominator would increase the rat population infection rate to Leptospira.  The epidemiological interpretation and impact by Health Ministry would be different.

3. Although Leptospira outbreaks were reported in the Sidney area before, and one human case was also reported, the rat finding in this study points out to push of more water, reservoirs, and febrile patient examinations in the hospitals. 

4. Leptospira serological and molecular surveillance of suspected dogs in the Sidney area, along with urban vertebrates such as opossums and local vertebrates, is urgently recommended after these results.

5. Antigenic variation of Leptospira or serovars is needed to prepare some vaccines and related preventive measures. Genomic sequencing would also be an excellent supplement for decision-makers.

6. The manuscript results are easily framed within the ONE HEALTH concept, which explains the importance of transit of human, animal, and wildlife interactions. A comment related to the discussion section would be welcome.

Author Response

Reviewer 2: Authors very well analyzed data from results; i.e., trapping methods, number of nights, qPCR, CT values, MAT serovar screening, etc.

Response: Thank you very much for this compliment about our manuscript.

Reviewer 2: I don't like the live rats' sample size; 16 versus 104 carcasses. I understand that working in the field is quite different from lab work. But, I would not be surprised if removing carcass numbers in the denominator would increase the rat population infection rate to Leptospira. The epidemiological interpretation and impact by Health Ministry would be different.

Response: We agree with this Reviewer’s comment, and the effect the different sampling methods may have had on our results. We do already report the difference in Leptospira prevalence between live-trapped rats and carcasses in the Results:

(L242-243)

‘In total, 4/16 (25%) live-trapped rats were qPCR positive, and 5/95 (5.3%) rat carcasses from killing stations were qPCR positive.’

And we do our best to address this possible limitation in the Discussion:

(L332-339)

‘Kidney samples from live-trapped rats were significantly more likely to test qPCR positive than rats collected from killing stations, likely due to sample DNA decomposition, considering up to 4 hours might have passed between death and carcass collection. Although live trapping is much more laborious than collecting carcasses, with daily trap checks re-quired for six consecutive days, future studies should focus on live rat trapping to mini-mise the possibility of false-negative qPCR results, facilitate MAT, and increase the likelihood of successful bacterial culture to identify the Leptospira serovar/s present.’

Reviewer 2: Although Leptospira outbreaks were reported in the Sydney area before, and one human case was also reported, the rat finding in this study points out to push of more water, reservoirs, and febrile patient examinations in the hospitals.

Response: Thank you for this excellent point. In response, we have added further detail in the Discussion and Conclusion.

(Discussion L368-371)

‘Other investigations of potential wildlife reservoirs of Leptospira infection, and environmental sources of Leptospira contamination (in particular water reservoirs), are also needed to better understand the risks of direct and indirect transmission of leptospirosis in Australia.’

(Conclusion L398-400)

‘Human physicians in Sydney, and elsewhere in NSW, should consider leptospirosis in at-risk patients presenting with symptoms possibly attributable to infection with leptospires.’

Reviewer 2: Leptospira serological and molecular surveillance of suspected dogs in the Sydney area, along with urban vertebrates such as opossums and local vertebrates, is urgently recommended after these results.

Response: Thank you, we have added a sentence in the Discussion to address this suggestion.

(Discussion L363-365)

‘Ongoing serological and molecular surveillance of possibly infected dogs and rats in the Sydney area, and bacterial culture for Leptospira organisms, is therefore important.’

Opossums are not present in Australia, and we already address the need for surveillance of local vertebrates including brushtail possums, kangaroos, and platypuses in the final paragraph of the Discussion.

Reviewer 2: Antigenic variation of Leptospira or serovars is needed to prepare some vaccines and related preventive measures. Genomic sequencing would also be an excellent supplement for decision-makers.

Response: Thank you, we have added a sentence in the Discussion to address this suggestion.

(Discussion L365-367)

‘Since vaccines to prevent leptospirosis are serovar-specific, identification of common serovars by culture and genomic sequencing will also assist with vaccine development and other related preventative measures.’

Reviewer 2: The manuscript results are easily framed within the ONE HEALTH concept, which explains the importance of transit of human, animal, and wildlife interactions. A comment related to the discussion section would be welcome.

Response: Thank you, we have added a sentence in the Discussion to address this suggestion, and ‘One Health’ has also been added as a keyword.

(L352-355)

‘This finding has potentially important implications for rat management interventions from a One Health perspective. One Health is an all-encompassing term that considers animal, human, and environmental health, and particularly the interplay between all three aspects [30].’

[30] Shaheen, M.N.F. The concept of one health applied to the problem of zoonotic diseases. Rev Med Virol. 2022, 32, 2326.

Reviewer 3 Report

This work aimed to estimate the prevalence of pathogenic Leptospira spp in the urban population of rats.

Abstract:

Among the results, the authors mention that the proportion of rats with Leptospira DNA in their kidneys was 9/111 (8.1%). qPCR tests of blood (n = 9) and urine (n = 4) samples were negative. None of the 10 serum samples tested positive for MAT. However, they conclude: “These findings suggest that rats in Sydney might play a role in transmitting leptospirosis to dogs and people.” Is that possible? Having only 8.1% of positive tests.

Methodology:

I consider that methodology is sound. However, the author does not use any methodology to confirm that what they amplified is what they expected, such as agarose gel electrophoresis or sequencing of the fragments to confirm. Although there were many samples, the positive ones were very few (8.1%). If it is possible, the authors would consider it.

 Line 165-166 - “The assay has a 50% limit of detection (LOD) of 0.026 fg/μL, which was used to determine a positive or negative result. A negative result was determined when no amplification occurred in the sample well, or when amplification was below the assigned LOD threshold for the assay (i.e., <0.026 fg/μL)” However, they do not mention or reference where that value or parameter comes from to determine if that PCR reaction is positive or negative.

Line 143-145 - At the University of Sydney (UoS), they found to detect a 242 bp fragment of the LipL32 gene belonging to pathogenic Leptospira species. Possibly for this study, it would have been convenient to carry out a conventional PCR since it is something more qualitative.

Line 158-160 - While at the Macarthur Agricultural Institute (EMAI), they found to detect an 87 bp fragment belonging to the rrs (16S) gene sequence of pathogenic Leptospira species. Although the qPCR methodology is adequate, it remains in doubt whether the size of the amplicon could play an important role when carrying out the assay.

Results:

Line 197-198 - The demographics of the 120 rats are shown in Error! Reference source not found.. ¿What is this?

Line 208-213 - In UoS ​​the proportion of rats with Leptospira was 9/111 (8.1%) with a positive Ct measurement of 34.4. In EMAI 6/111 (5.4%) of rats were positive for Leptospira with a Ct of 26.8.

What is this difference in Ct due to? (Not discussed). Also, in line 213 it mentions: “Five of these six qPCR positive kidney samples were also qPCR positive at the use.” Despite being a different gene, both look for the same thing (pathogenic Leptospira species) then; In theory, the same positive samples from the UoS should be the same from the EMAI. Why is this not the case?

Discussion:

They performed microscopic agglutination tests (MAT) on serum samples to identify the different Leptospira serotypes, a study in which no positive tests were obtained. This is not discussed.

 Minor editing of English language required.

Author Response

Abstract:

Among the results, the authors mention that the proportion of rats with Leptospira DNA in their kidneys was 9/111 (8.1%). qPCR tests of blood (n = 9) and urine (n = 4) samples were negative. None of the 10 serum samples tested positive for MAT. However, they conclude: “These findings suggest that rats in Sydney might play a role in transmitting leptospirosis to dogs and people.” Is that possible? Having only 8.1% of positive tests.

Response: We agree with this Reviewer’s comment that the findings are not conclusive. For this reason in the Abstract we deliberately say ‘might play a role’, and in the Discussion we further qualify this by saying:

(L283-286)

‘Although the viability of the Leptospira DNA material detected was not studied, and direct transmission was not demonstrable, these findings suggest that rats in Sydney might play a role in the transmission of leptospirosis to dogs and people.’

Reviewer 3:

Methodology:

I consider that methodology is sound. However, the author does not use any methodology to confirm that what they amplified is what they expected, such as agarose gel electrophoresis or sequencing of the fragments to confirm. Although there were many samples, the positive ones were very few (8.1%). If it is possible, the authors would consider it.

Response: Thank you for this observation. However, we respectfully disagree with the Reviewer regarding the need for an additional methodology to confirm that the products we amplified corresponded to the fragment we were looking for. We included spiked and unspiked uninfected pathogen-free rat samples, as well as positive (5 ng of the synthesised fragment) and negative controls (blank) without rat DNA each run. Moreover, the fragment targeted in our methodology have been shown to have high sensitivity, with a Lower Limit of Detection (LLOD) of at least 0.013 fg/µL (50 GE/µL), and high specificity. Thus, we consider further confirmation of the fragment amplified, although comprehensive, unnecessary. In response to this Reviewer’s comment, however, we have now added further details relevant to this in our methods

(Lines 155-163)

‘This has been shown to be high sensitivity and specificity, with a lower limit of detection (LLOD) of 0.013 fg/µL [3]. DNA extracted from a pathogen-free male R. norvegicus laboratory rat (Wistar) was used to create positive (spiked) and negative controls (unspiked). Positive controls were spiked with 5 ng of the synthesised known 242 bp fragment, and negative controls were unspiked. Additionally, a positive control consisting of only 5 ng of the synthesised known 242 bp fragment and a blank were also included in each run. Controls were included in triplicate to form a standard curve and determine appropriate detection thresholds.’

Reviewer 3:

Line 165-166 - “The assay has a 50% limit of detection (LOD) of 0.026 fg/μL, which was used to determine a positive or negative result. A negative result was determined when no amplification occurred in the sample well, or when amplification was below the assigned LOD threshold for the assay (i.e., <0.026 fg/μL)” However, they do not mention or reference where that value or parameter comes from to determine if that PCR reaction is positive or negative.

Response: The Limit Of Detection (LOD) was defined as the limit where 50% of qPCRs were successful. This was estimated using 8 replicates of Leptospira plasmid DNA at 20, 2, 0.2, 0.02, 0.002 and 0.0002 fg/μL. The LOD was calculated using GraphPad Prism software with a nonlinear regression model using a sigmoidal dose-response, which determined the LOD for this assay was 0.026 fg/μL.

This detail has been added to the manuscript:

(L173-176)

‘This LOD was determined using eight replicates of Leptospira plasmid DNA at 20, 2, 0.2, 0.02, 0.002 and 0.0002 fg/μL, and calculated using GraphPad Prism software (San Diego, CA, USA) with a nonlinear regression model using a sigmoidal dose-response.’

Reviewer 3:

Line 143-145 - At the University of Sydney (UoS), they found to detect a 242 bp fragment of the LipL32 gene belonging to pathogenic Leptospira species. Possibly for this study, it would have been convenient to carry out a conventional PCR since it is something more qualitative.

Response: We agree with this Reviewer that given the small number of samples tested at the University of Sydney (UoS), a conventional PCR would have been adequate. However, conventional PCR would have required further confirmation of amplification by agarose gel electrophoresis or sequencing. Furthermore, the way this project was initially planned included testing a considerable number of samples at the UoS, to which real-time PCR is better suited. Unfortunately, the low number of samples tested at UoS was a consequence of the low trapping success rate.

Reviewer 3:

Line 158-160 - While at the Macarthur Agricultural Institute (EMAI), they found to detect an 87 bp fragment belonging to the rrs (16S) gene sequence of pathogenic Leptospira species. Although the qPCR methodology is adequate, it remains in doubt whether the size of the amplicon could play an important role when carrying out the assay.

Response: The EMAI Leptospira PCR assay is based on a published assay (Smythe et al., 2002, A Quantitative PCR (TaqMan) assay for pathogenic Leptospira spp, BMC Infectious Diseases, 2:13) and is specific for pathogenic Leptospira spp. This was confirmed at EMAI before using the assay via BLASTn of the target sequence to confirm the specificity of the assay as stated in the paper. This detail has also been added to the Methods section:

(L166-167)

‘The specificity of the assay was confirmed at EMAI prior to testing by NCBI BLASTN of the target sequence.’

Reviewer 3:

Results:

Line 197-198 - The demographics of the 120 rats are shown in Error! Reference source not found.. ¿What is this?

Response: We are uncertain about what the Reviewer is referring to with this comment? We suspect it might have been a broken hyperlink produced in our manuscript during submission, but we currently cannot see the issue in our revised manuscript.

Reviewer 3:

Line 208-213 - In UoS ​​the proportion of rats with Leptospira was 9/111 (8.1%) with a positive Ct measurement of 34.4. In EMAI 6/111 (5.4%) of rats were positive for Leptospira with a Ct of 26.8.

What is this difference in Ct due to? (Not discussed). Also, in line 213 it mentions: “Five of these six qPCR positive kidney samples were also qPCR positive at the UoS.” Despite being a different gene, both look for the same thing (pathogenic Leptospira species) then; In theory, the same positive samples from the UoS should be the same from the EMAI. Why is this not the case?

Response: We appreciate the Reviewer’s observation. In reference to the proportion of rats we detected Leptospira at the UoS vs EMAI, we have changed the wording. Only 28 rats were tested at UoS not 111, so the proportion from UoS is 21%. In reference to the differences in Ct, both assays used differed on the location where they were performed, the fragment targeted, reactants, cycling conditions, and the proportion of samples tested from live caught animals vs carcasses, all factors known to affect Ct. Thus, comparing Ct values between the two assays is meaningless. Finally, the difference in the number of rats with Leptospira DNA detected at UoS vs EMAI may have been due to several factors, including differences in sensitivity and specificity between assays. We have now included further details in the discussion to address this.

(L339-343)

‘In general, we found that the qPCR assays used (UoS and EMAI), were highly congruent, with only one sample classified as positive at EMAI but negative at the UoS, and one sample classified as negative at EMAI but positive at the UoS. These discordant results might have been due to differences in the sensitivities of the two qPCR assays [3,21]‘.

Reviewer 3:

Discussion:

They performed microscopic agglutination tests (MAT) on serum samples to identify the different Leptospira serotypes, a study in which no positive tests were obtained. This is not discussed.

Response: We do briefly address this finding in the Discussion:

‘Serum from only 10/16 live trapped rats was able to be obtained, and all ten rats that were MAT negative were also qPCR negative for Leptospira with kidney testing… Although live trapping is much more laborious than collecting carcasses, with daily trap checks required for six consecutive days, future studies should focus on live rat trapping to minimise the possibility of false-negative qPCR results, facilitate MAT, and increase the likelihood of successful bacterial culture to identify the Leptospira serovar/s present.’

We felt this was sufficient to address the negative MAT results, and that it was better to focus our attention in the Discussion on the positive qPCR results and the possible implications of this finding.

Reviewer 4 Report

Dear Authors,

The manuscript "Pathogenic Leptospira species in urban rats in Sidney, Australia" describes the detection of pathogenic Leptospira in samples in rodents, the main reservoir of Leptospira. Some comments below.

Introduction

Line 61: MAT results is in serogroup. “…which three dogs (50%) had titres for Icterohaemorrhagiae serogroup”.

Line 68-70.: MAT result is in serogroup.

Line 81: example of serovars.

Methods

Is there IACUC protocol number?

Line 144: lipL32 is in italic.

Line 136-175: did you include internal positive control? This protocol used it. The samples were frozen. What is the low limit detection using frozen samples? Did you do spike to determine the low limit detection in kidney, urine, and blood samples? Did you include Standard Curve? If yes, explain how you did it.

Line 158-177: 16s qPCR, is specific for pathogenic? It is confused about LOD. Did you use clinical samples to determine LOD? Did you include Standard Curve? If yes, explain how you did it.

Line 178-183: MAT is serogroup results, include the serogroups as X? different Leptospira serogroups: Ballum (serovar Arborea) …

Line 194: suggestion to change the title of 3.1. “Rodent survey”.

Results

Line 208: Does results include both PCR? It is confused.

Author Response

Introduction

Line 61: MAT results is in serogroup. “…which three dogs (50%) had titres for Icterohaemorrhagiae serogroup”.

Response: As this Reviewer correctly points out, Leptospira interrogans serovar Copenhageni and serovar Icterohaemorrhagiae belong to the same serogroup and share common antigens, which cross-react serologically when using the MAT. However, The WHO/FAO/OIE Collaborating Centre for Reference & Research on Leptospirosis (that did the testing for the study referenced) only used serovar Copenhageni in their MAT panel as the serological representative for Australia (i.e., serovar Icterohaemorrhagiae was not used). Therefore, we believe what we have written (‘of which three dogs (50%) had titres against L. interrogans serovar Copenhageni’) is correct.

Reviewer 4:

Line 68-70: MAT result is in serogroup.

Response: Same response as above (testing performed at the same WHO laboratory).

This was not the case for the MAT performed in the current study, and therefore the description for MAT in the methods has been changed accordingly (see below).

Reviewer 4:

Line 81: example of serovars.

Response: ‘including serovar Zanoni’ has been added, based on results from the study referenced [9]. Due to the scarcity of information on reports of leptospirosis in vaccinated dogs in Australia, we hope this comment is also addressed by the following sentences in the Introduction:

(L84-87)

‘At least one fatal case of canine leptospirosis has been reported in a dog vaccinated with Protech® C2i nine months prior. However, the causative serovar could not be determined, likely due to insufficient time for seroconversion [15].’

[9] Orr, B.; Westman, M.E.; Malik, R.; Purdie, A.; Craig, S.B.; Norris, J.M. Leptospirosis is an emerging infectious disease of pig-hunting dogs and humans in North Queensland. PLOS Neglected Tropical Diseases 2022, 16, e0010100, doi:10.1371/journal.pntd.0010100.

[15] Griebsch, C.; Kirkwood, N.; Ward, M.; So, W.; Weerakoon, L.; Donahoe, S.; Norris, J. Emerging leptospirosis in urban Sydney dogs: a case series (2017–2020). Australian Veterinary Journal 2022, 100, 190-200, doi: 10.1111/avj.13148.

Reviewer 4:

Methods

Is there IACUC protocol number?

Response: Australia does not use IACUC numbers. Instead, research is covered by Australian Research Authorities (ARA). This is covered by the Institutional Review Board Statement.

Institutional Review Board Statement: Animal ethics approval for rat sampling was granted by the University of Sydney Animal Ethics Committee (approval number 2020/1725). N00/1-2013/3/5920, 2017/1130 and 2019/1665).

Reviewer 4:

Line 144: lipL32 is in italic.

Response: Thank you, corrected.

Reviewer 4:

Line 136-175: did you include internal positive control? This protocol used it. The samples were frozen. What is the low limit detection using frozen samples? Did you do spike to determine the low limit detection in kidney, urine, and blood samples? Did you include Standard Curve? If yes, explain how you did it.

Response: We appreciate the reviewer’s observation. We have added further details to this section.

(L155-163)

‘DNA extracted from a pathogen-free male R. norvegicus laboratory rat (Wistar) was used to create positive (spiked) and negative controls (unspiked). Positive controls were spiked with 5 ng of the synthesised known 242 bp fragment, and negative controls were unspiked. Additionally, a positive control consisting of only 5 ng of the synthesised known 242 bp fragment and a blank were also included in each run. Controls were included in triplicate to form a standard curve and determine appropriate detection thresholds.’

Reviewer 4:

Line 158-177: 16s qPCR, is specific for pathogenic? It is confused about LOD. Did you use clinical samples to determine LOD? Did you include Standard Curve? If yes, explain how you did it.

Response: Yes, the PCR assay used is specific for pathogenic Leptospira spp. From reference [21]:

‘A detectable product was observed for all the 23 pathogenic leptospiral strains. Other bacterial strains and non-pathogenic leptospiral strains did not yield amplification products.’

[21] Smythe, L.D.; Smith, I.L.; Smith, G.A.; Dohnt, M.F.; Symonds, M.L.; Barnett, L.J.; McKay, D.B. A quantitative PCR (TaqMan) assay for pathogenic Leptospira spp. BMC Infectious Diseases 2002, 2, 1-7.

The specificity of the assay was confirmed at EMAI prior to testing by NCBI BLASTN of the target sequence to confirm the specificity of the assay as stated in the paper. Clinical samples were not used to determine limit of detection (LOD). The LOD was defined as the limit where 50% of qPCRs were successful. This was estimated using 8 replicates of Leptospira plasmid DNA at 20 fg/μL, 2 fg/μL, 0.2 fg/μL, 0.02 fg/μL, 0.002 fg/μL and 0.0002 fg/μL. The LOD was calculated using GraphPad Prism software with a nonlinear regression model using a sigmoidal dose-response, which determined the LOD for this assay was 0.026 fg/μL.

A standard curve was used for all qPCR runs. This standard curve was created by running four known plasmid standard concentrations from a dilution series in duplicate for each run to ensure the run had performed within specific acceptance criteria (R2 > 0.98, and Efficiency = 85%-110%) and if the standard curve values produced fell outside of these criteria, then it was deemed to have failed and needed to be re-run.

These details have all been added to the Methods section:

(L173-176)

‘This LOD was determined using eight replicates of Leptospira plasmid DNA at 20, 2, 0.2, 0.02, 0.002 and 0.0002 fg/μL, and calculated using GraphPad Prism software (San Diego, CA, USA) with a nonlinear regression model using a sigmoidal dose-response.’

(L182-186)

‘A standard curve was created by running four known plasmid standard concentrations from a dilution series in duplicate, with acceptable run criteria defined as R2 > 0.98 and efficiency = 85%-110%. If the standard curve values fell outside of these criteria, the run was deemed to have failed and testing was repeated.’

Reviewer 4:

Line 178-183: MAT is serogroup results, include the serogroups as X? different Leptospira serogroups: Ballum (serovar Arborea) …

Response: Thank you for this excellent suggestion. The methods have been changed accordingly.

(L194-200)

‘This included testing against 10 different Leptospira serovars (sv.), pertaining to 9 different Leptospira serogroups: Serogroup Ballum (sv. Arborea), serogroup Australis (sv. Australis), serogroup Canicola (sv. Canicola), serogroup Icterohaemorrhagiae (sv. Copenhageni and sv. Icterohaemorrhagiae), serogroup Grippotyphosa (sv. Grippotyphosa), serogroup Sejroe (sv. Hardjo), serogroup Pomona (sv. Pomona), serogroup Tarassovi (sv. Tarassovi) and serogroup Pyrogenes (sv. Zanoni).’

The Abstract and Discussion has also been updated accordingly:

(Abstract L18-21)

‘A subset of rats also had qPCR testing performed on whole blood and urine, and Microscopic Agglutination Testing (MAT) that included a panel of 10 Leptospira serovars from 9 different Leptospira serogroups was performed on a subset of serum samples.’

(Discussion L328-329)

‘MAT of a subset of samples was unfortunately unable to identify the Leptospira serogroups present in rats in the current study.’

Reviewer 4:

Line 194: suggestion to change the title of 3.1. “Rodent survey”.

Response: Thank you for this suggestion, changed as suggested.

Reviewer 4:

Results

Line 208: Does results include both PCR? It is confused.

Response: Thank you, this has been reworded to address this confusion.

(L226-227)

‘Based on combined results from qPCR testing at both sites, the proportion of rats carrying Leptospira was 9/111 (8.1%).’

Round 2

Reviewer 3 Report

All my concerns were considered in the final version. I have no more comments.

Author Response

Thank you for your review and the endorsement to publish.

Reviewer 4 Report

 1) we believe what we have written (‘of which three dogs (50%) had titres against L. interrogans serovar Copenhageni’) is correct. It is not correct. The MAT results is in serogroup. Should change for Icterohaemorrhagiae serogroup.

2) When you determine the LLOD, you should use the sample you are testing. It is not clear if you did the spike using the clinical sample. Did you only dilution the DNA?

Author Response

Response to Reviewer 4:

Reviewer 4:

1) we believe what we have written (‘of which three dogs (50%) had titres against L. interrogans serovar Copenhageni’) is correct. It is not correct. The MAT results is in serogroup. Should change for Icterohaemorrhagiae serogroup.

Response: Thank you, this has now been corrected.

(L60-62)

‘A 2008 Australian serosurvey of dogs reported six seropositive dogs from a Sydney shelter (Yagoona), of which three dogs (50%) had titres against L. interrogans serogroup Icterohaemorrhagiae, and a local rat problem was hypothesised [13].’

2) When you determine the LLOD, you should use the sample you are testing. It is not clear if you did the spike using the clinical sample. Did you only dilution the DNA?

Response: Thank you, we have clarified the methods section to clarify this point. Initial PCR testing at the first location (the University of Sydney) did not have a lower limit of detection (LOD) determined for the assay, and the LOD reported was for the published assay. Repeat PCR testing of all samples at the second location (Elizabeth Macarthur Agricultural Institute), however, did have a lower LOD determined using plasmid DNA generated from a positive field sample received at EMAI and confirmed by NCBI BLASTn of the target sequence.

We hope the below edits address this confusion.

(L155-166 – PCR testing at the University of Sydney [19])

‘This has been shown to be highly sensitive and specific, with a lower limit of detection (LOD) of 0.013 fg/µL [19]. DNA extracted from a pathogen-free male R. norvegicus laboratory rat (Wistar) was used to create positive (spiked) and negative controls (unspiked). Positive controls were spiked with 5 ng of the synthesised known 242 bp fragment, and negative controls were unspiked. Additionally, a positive control consisting of only 5 ng of the synthesised known 242 bp fragment and a blank were also included in each run. Controls were included in triplicate to determine if rat genomic DNA would interfere with the detection, either by blocking the amplification of the fragment or by erroneous amplification of rat DNA by the probes used. Neither of those cases was detected in our assay, with comparable Ct levels for both types of positive (i.e., spiked and fragment) and negative controls (i.e., unspiked and blank).’

  1. Stoddard, R.A.; Gee, J.E.; Wilkins, P.P.; McCaustland, K.; Hoffmaster, A.R. Detection of pathogenic Leptospira spp. through TaqMan polymerase chain reaction targeting the LipL32 gene. Diagn Microbiol Infect Dis 2009, 64, 247-255, doi: 10.1016/j.diagmicrobio.2009.03.014.

(L174-191 – PCR testing at EMAI [20])

‘An in-house designed Leptospira plasmid standard (the 87 bp rrs target from a positive field sample received at EMAI and cloned into a pCR2.1 TOPO™ vector, ThermoFisher, Waltham, MA, USA) was created and used for all qPCR runs as a positive control and to form a standard curve to quantify results. The correct sequence of the 87 bp plasmid insert was confirmed at EMAI prior to testing by NCBI BLASTn of the target. Eight replicates of the Leptospira plasmid DNA at 20, 2, 0.2, 0.02, 0.002 and 0.0002 fg/μL were used to calculate a LOD using GraphPad Prism software (San Diego, CA, USA) with a nonlinear regression model using a sigmoidal dose-response. The 50% LOD for the assay was determined to be 0.026 fg/µL. This LOD was used to determine a positive or negative result. A positive result was determined when amplification in the sample well was above the LOD. A negative result was determined when no amplification occurred in the sample well, or when amplification was below the LOD (i.e., < 0.026 fg/µL). A standard curve was created by running four known plasmid standard concentrations from a dilution series in duplicate, with acceptable run criteria defined as R2 > 0.98 and efficiency = 85%-110%. If the standard curve values fell outside of these criteria, the run was deemed to have failed and testing was repeated. Three negative controls (PCR reagents without template DNA added, PCR reagents with MGW added instead of template DNA, and an extraction process control) were also included in each PCR run.’

  1. Smythe, L.D.; Smith, I.L.; Smith, G.A.; Dohnt, M.F.; Symonds, M.L.; Barnett, L.J.; McKay, D.B. A quantitative PCR (TaqMan) assay for pathogenic Leptospira spp. BMC Infectious Diseases 2002, 2, 1-7.
